# From Thousands to Billions: 3D Visual Language Grounding via Render-Supervised Distillation from 2D VLMs

Ang Cao [1 2]   Sergio Arnaud [2]   Oleksandr Maksymets [2]   Jianing Yang [1 2]   Ayush Jain [2 3]   Ada Martin [2]
Vincent-Pierre Berges [2]   Paul McVay [2]   Ruslan Partsey [2]   Aravind Rajeswaran [2]   Franziska Meier [2]
Justin Johnson [1]   Jeong Joon Park [1]   Alexander Sax [2]

## Abstract

3D vision-language grounding faces a fundamental data bottleneck: while 2D models train on billions of images, 3D models have access to only thousands of labeled scenes–a six-order-of-magnitude gap that severely limits performance. We introduce *LIFT-GS*, a practical distillation technique that overcomes this limitation by using differentiable rendering to bridge 3D and 2D supervision. LIFT-GS predicts 3D Gaussian representations from point clouds and uses them to render predicted language-conditioned 3D masks into 2D views, enabling supervision from 2D foundation models (SAM, CLIP, LLaMA) without requiring any 3D annotations. This render-supervised formulation enables end-to-end training of complete encoder-decoder architectures and is inherently model-agnostic. LIFT-GS achieves state-of-the-art results with 25.7% mAP on open-vocabulary instance segmentation (vs. 20.2% prior SOTA) and consistent 10-30% improvements on referential grounding tasks. Remarkably, pretraining effectively multiplies fine-tuning datasets by 2×, demonstrating strong scaling properties that suggest 3D VLG currently operates in a severely data-scarce regime. Project page: https://liftgs.github.io.

## 1. Introduction

When a user mentions *the keys by the door* or *the blue mug on the table*, they use language to indicate a specific set

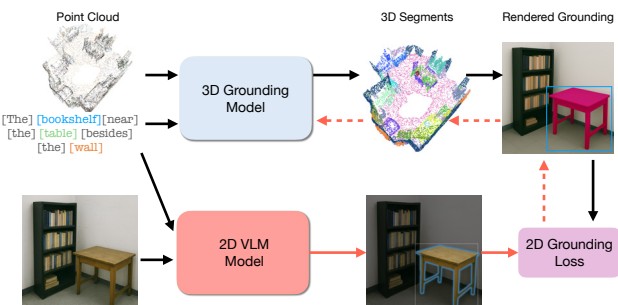

**Figure 1: LIFT-GS Overview.** We train a powerful 3D vision language grounding model (*i.e.*, 3D mask decoder) with point clouds and language as inputs by learning from 2D VLM foundation models without any 3D supervision.

of objects and 3D locations in space. Such *3D language grounding* provides a particularly natural interface for people to communicate about their surroundings. For AI systems operating in physical spaces, identifying the 3D masks or bounding boxes indexed by language queries represents a core functionality, with applications across autonomous navigation, robotic manipulation, and AR/VR.

Yet despite its importance, *3D vision-language grounding* (3D VLG) faces a fundamental bottleneck: data scarcity. While 2D vision-language models are trained on billions of labeled images and masks (Achiam et al., 2023; Touvron et al., 2023; Radford et al., 2021; Labs, 2023), existing 3D VLG models have access to only thousands of labeled 3D scenes and masks. This six-order-of-magnitude gap in data availability severely limits the capabilities of current 3D grounding systems, creating one of the most significant challenges in embodied AI.

A common workaround to this scarcity constructs 3D feature fields from 2D features (*e.g.*, CLIP embeddings) and performs text queries via dot products between the text and 3D embeddings. Although this provides good generalization, performance degrades with more detailed descriptions typical of real-world queries, as illustrated in Figure 3. From this perspective, the dual-encoder approach falls short of *3D grounding* as it contradicts a core grounding requirement.

In this paper, we ask: can we combine the best part of both

This work was partially done during the internship of Ang Cao at Meta. [1]University of Michigan, Ann Arbor [2]Fundamental AI Research (FAIR), Meta [3]Carnegie Mellon University. Correspondence to: Ang Cao <ancao@umich.edu>, Alexander Sax <ssax@meta.com>.

pipelines, *i.e.*, training a powerful grounding model while still overcoming the data scarcity by learning from powerful 2D models? The key insight is that differentiable rendering provides a natural bridge between 3D and 2D. If we can predict 3D masks and render them into 2D views, we can supervise them using 2D foundation models that have been trained on internet-scale data. This approach could enable training 3D models without any 3D mask annotations.

We introduce *Language-Indexed Field Transfer with Gaussian Splatting* (LIFT-GS), which implements this idea as a practical training pipeline. Given a point cloud and language query, LIFT-GS predicts 3D Gaussian representations that can be rendered into multiple 2D views. These rendered masks are then supervised using pseudo-labels from 2D foundation models. We show results where SAM provides mask supervision (Kirillov et al., 2023), and CLIP or LLaMa provide language understanding (Radford et al., 2021; Meta AI, 2024). The approach effectively distills internet-scale 2D knowledge into 3D understanding.

This render-supervised formulation offers several key advantages. First, it is inherently architecture-agnostic; specifying only the outputs leaves flexibility in underlying model design. Second, this allows us to overcome fundamental scaling limitations by training a large transformer decoder instead of previous dual-encoder approaches (as shown in Fig 3) (Zhu et al., 2023b; Gu et al., 2024). Third, the approach is highly practical: LIFT-GS operates directly on raw point clouds from sensors, such as the outputs from SLAM or SfM systems, eliminating the preprocessing and feature fusion required by methods like ConceptFusion (Jatavallabhula et al., 2023b; Arnaud et al., 2025), reducing inference time from 60 seconds to just 1 second.

Our experiments validate the effectiveness of this approach. LIFT-GS achieves state-of-the-art performance on standard 3D VLG benchmarks, with 25.7% mAP on open-vocabulary instance segmentation (vs. 20.2% previous SOTA) and consistent 10-30% relative improvements on referential grounding tasks. More importantly, we observe that across data scales for SFT, pretraining effectively "multiplies" the fine-tuning dataset by approximately a constant factor (2x). That is, a pretrained model with 50% of fine-tuning data matches the performance of training from scratch with 100% data. This somewhat counterintuitive observation indeed matches empirical data scaling laws for pretraining in other modalities (Hernandez et al., 2021), and the fact that this scaling coefficient remains constant without diminishing returns across data scales adds an empirical data point that 3D VLG currently operates in the very low-data regime [1].

The implications extend beyond 3D grounding. Render su-

---

[1](Hernandez et al., 2021) define a "low-data regime as having 10% or less of the amount of data it would take to get to 99% of the performance that infinite data would yield."

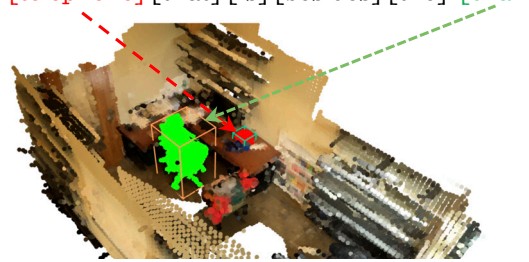

[The] [telephone] [that] [is] [besides] [the] [chair]

Figure 2: **3D Referential Grounding.** For each mentioned instance in a text description, predict a 3D mask and map it to corresponding text tokens.

pervision is powerful, but we demonstrate that it can serve as a bridge for large-scale knowledge transfer from 2D foundation models to 3D models. Any 3D/4D task with renderable outputs can potentially leverage 2D supervision. As 2D foundation models continue to improve and scale, 3D models trained using render-supervised distillation are positioned to benefit. This opens the possibility of training 3D understanding models at the scale of 2D datasets–which would represent a fundamental shift from the current paradigm of limited 3D annotations.

To summarize, our contributions are:

- **A render-supervised training pipeline for 3D vision-language grounding that requires only 2D supervision.** We show how differentiable rendering enables training 3D models with 2D losses, eliminating dependence on scarce 3D annotations.
- **Demonstrating a pseudo-labeling strategy for distilling 2D foundation models into 3D.** LIFT-GS shows using SAM, CLIP, and LLMs to generate 2D supervision.
- **State-of-the-art performance in realistic evaluations.** LIFT-GS achieves SOTA results using sensor point clouds common in embodied settings, with detailed ablations revealing scaling properties.

## 2. Related Work

### 2.1. The Data Scarcity Challenge in 3D VLG

3D Vision-Language Grounding (3D VLG) maps language descriptions to corresponding 3D masks or bounding boxes in observed scenes (Yuan et al., 2021; Roh et al., 2021; Yang et al., 2021). Despite its fundamental importance for embodied AI, existing 3D VLG datasets contain only thousands of annotated scenes (Dai et al., 2017; Yeshwanth et al., 2023) compared to billions of images used for training large multimodal models (Meta AI, 2024).

This scarcity stems from prohibitive annotation costs. Creating 3D instance masks requires minutes per example even with assisted tools (Dai et al., 2017), and human verification remains necessary (Arnaud et al., 2025; Majumdar et al.,

2024). All existing 3D VLG methods require ground-truth 3D masks or bounding boxes during training (Yuan et al., 2021; Zhu et al., 2023c; 2024; Zhang et al., 2024), with many also requiring them during inference (Fang et al., 2024; Zhang et al., 2023b). This creates a fundamental scaling barrier–these approaches cannot leverage the vast 2D data that has driven progress in 2D understanding.

## 2.2. Bridging 2D and 3D: From Lifting to Learning

**2D-to-3D *Lifting* via Optimization.** Recent work addresses data scarcity by lifting 2D models to 3D through per-scene optimization. Methods use depth unprojection with heuristic merging (*e.g.*, voxel voting) (Jatavallabhula et al., 2023a; Zhou et al., 2024; Xu et al., 2023b) or differentiable rendering to optimize 3D representations matching 2D features (Kerr et al., 2023; Kim et al., 2024; Gu et al., 2024) or masks (Cen et al., 2023; Xu et al., 2023a). While these leverage 2D foundation models, they suffer from: (1) slow optimization (minutes per scene), (2) accumulated errors from reconstruction and merging, and (3) inability to improve with more data. The fixed lifting pipelines may be a bottleneck that explains why LIFT-GS outperforms models trained on these lifted pseudolabels (Genova et al., 2021; Peng et al., 2023).

**Render-Supervised *Learning*.** Differentiable rendering enables training 3D models directly by rendering their predictions into 2D and supervising via 2D losses. While initially used for reconstruction (Hong et al., 2024; Tang et al., 2024; Cao et al., 2024; Szymanowicz et al., 2025), (Irshad et al., 2024; Zhu et al., 2023b) use it to representation learning, like PonderV2 adds CLIP losses on rendered pixels (Zhu et al., 2023a). However, existing 3D VLG methods have significant limitations: PonderV2 only trains encoders and relies on ground-truth category labels, while other methods apply simple photometric losses without leveraging 2D VLMs. LIFT-GS overcomes these challenges by extending render-supervision in two key ways: (1) it enables training unified encoder-decoder architectures for 3D VLG tasks, and (2) it performs knowledge distillation from 2D foundation models (VLMs, SAM, CLIP) through pseudo-labeling, removing the need for 3D annotations during pretraining.

## 2.3. Architecture: From Dot-Products to Joint Attention

**Limitations of Dual-Encoder Methods**. Almost all prior 3D grounding approaches that distill from 2D vision-language models eventually compute the mask as the dot-product similarity between 3D features and text embeddings (Radford et al., 2021; Guo et al., 2024; Qin et al., 2023; Kerr et al., 2023; Peng et al., 2023). When modality embeddings are computed independently, as is the case in CLIP, these "dual-encoder" models behave as bag-of-words systems (Yuksekgonul et al., 2022). Using these encoders

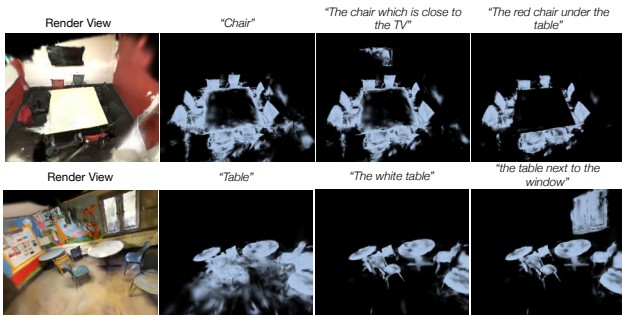

Figure 3: **3D grounding with CLIP-style (dual-decoder) method.** Grounding heatmaps from a representative approach (Guo et al., 2024). Heatmaps are computed using dot product similarity between visual tokens and text tokens (as in the CLIP objective), encoded independently. This performs effectively with very short prompts, but fails with more detailed queries, as shown in the image. LIFT-GS addresses this by jointly predicting tokens using a transformer decoder with expressive attention masks (see Figure 6 and experiments).

causes 3D models to inherit these fundamental limitations; as shown in the limited ability of 3D dual-encoder models to handle relational language crucial for referential grounding (e.g., "the chair next to the table") (Fig. 3).

**Multimodal Decoders**. Transformer decoders address the bag-of-words behavior by jointly processing the modalities together through learned attention mechanisms that enable proper handling of spatial relationships and multi-object references. Following large multimodal language models (LLaMA 3, GPT-4o, and Qwen 2.5), as well as recent 3D VLG SotA (Kamath et al., 2021; Jain et al., 2025; Arnaud et al., 2025), LIFT-GS employs a decoder-based architecture. LIFT-GS introduces the grounding loss described in Sec. 3.2 to train the decoder.

## 2.4. Foundation Model Distillation at Scale

Recent work on scaling up pseudolabeling pipelines shows that although 2D foundation models are increasingly capable (Hong et al., 2023; Arnaud et al., 2025), they currently exhibit significant limitations in spatial understanding over multiple frames, frequently hallucinating 3D spatial relationships (Majumdar et al., 2024; Yang et al., 2024). This is why LIFT-GS uses pseudolabels for pretraining: just as LLMs require supervised fine-tuning (SFT) to align noisy internet text with desired behaviors, LIFT-GS leverages noisy pseudolabels for large-scale pretraining, then uses 3D VLG SFT for state-of-the-art performance.

Our scaling analysis reveals suggests that even imperfect 2D spatial understanding generates meaningful training signal (multiplying fine-tuning data effectiveness by 2× in our experiments). As 2D models advance in spatial reasoning, this transfer benefit should amplify, potentially reducing reliance on 3D annotations and moving toward more zero-shot spatial understanding.

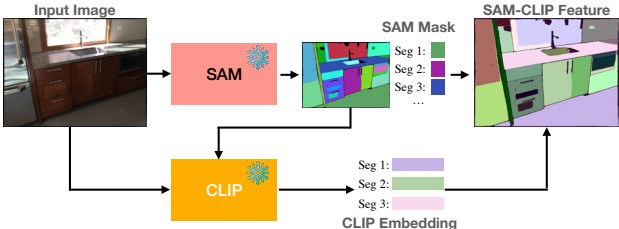

Figure 4: **SAM-CLIP Pseudo-Label Generation.** We leverage powerful 2D foundation models to generate *pseudo language queries*, i.e., CLIP embeddings, along with their corresponding ground-truth 2D masks for training. All pixels within the same mask share the same features.

This positions our approach at the intersection of three key trends: (1) the shift from optimization-based to learning-based 3D understanding, (2) the move from dual-encoder to multimodal decoder architectures for complex language grounding, and (3) the emergence of foundation model distillation as a solution to 3D data scarcity. Our experiments demonstrate that this approach not only achieves state-of-the-art performance but also exhibits strong scaling properties, suggesting significant potential for using cross-scene render-supervised distillation with improved multimodal foundation models and in other settings in besides 3D VLG.

## 3. Method

LIFT-GS is a (pre-)training pipeline for 3D VLG without 3D GT annotation. During training, it renders the predicted 3D masks from the target viewpoints for 2D supervision; during testing, 3D masks are used as outputs. This section covers the 3D VLG formalism in Sec.3.1, 2D loss used for supervision in Sec.3.2, pseudolabel generation in Sec.3.4, and implementation details in Sec.3.3.

### 3.1. Task Formulation

#### 3.1.1. *3D Vision-Language Grounding*

Figure 2 shows an example using a point cloud input ($\mathbf{P}$) and the text query ($\mathbf{Q}$) *"the black chair close to the table near the wall"*. Following MDETR (Kamath et al., 2021) and (Jain et al., 2025), LIFT-GS outputs a set of ($m = 256$) 3D mask candidates $\mathbf{M}$, and the correspondence matrix $\mathbf{C}$.

$$\text{3D VLG: } (\mathbf{P}, \mathbf{Q}) \mapsto (\mathbf{M}, \mathbf{C}), \quad (1)$$

**Matrix $\mathbf{C} \in \mathbb{R}^{m \times |Q|}$**: indicates the correspondence (*i.e.* probability logits) between $m$ 3D mask candidates and $|Q|$ text tokens, enabling the mapping of each text query to its most probable mask candidate based on the logits.

**3D Mask $\mathbf{M} \in \mathbb{R}^{m \times N}$**: stores mask logits over $N$ Gaussian primitives for $m$ 3D mask candidates. Each logit $\mathbf{M}_{i,j}$ represents the probability (after applying sigmoid) that the $j$-th Gaussian primitive is included in the $i$-th mask candidate.

This dual design elegantly handles two key challenges: (1) text tokens can refer to multiple instances (*e.g.*, "the **chairs**"), and (2) instances can be referenced multiple times with different descriptions throughout the text (*e.g.* both "chair" and "it"). It also extends readily to segmenting other modalities by adding more mappings.

The pointcloud has $|P|$ points and is of shape $|P| \times 6$ (XYZ + RGB), and the query token embeddings and are a matrix of size: $\mathbf{Q} \in \mathbb{R}^{|Q| \times F_Q}$. The text mapping $\mathbf{C}$ and Gaussian/point cloud mapping $\mathbf{M}$ are shown, with each instances highlighted in a different color in Figure 2.

#### 3.1.2. *3D VLG with Gaussian Masks $\mathbf{M}$*

Existing 3D VLG methods are limited by costly point cloud annotation, where annotation for point cloud masks is by far the costliest and slowest step in 3D VLG data collection. This is because human annotators must carefully segment the mask using a brush or assisted tool, which takes on the order of minutes per example (Dai et al., 2017). Even with model assistance, human verification is required in practice (Arnaud et al., 2025; Majumdar et al., 2024). This restricts training to only thousands of scenes, making data scarcity the key bottleneck.

LIFT-GS addresses this by leveraging differentiable rendering, distilling knowledge from multimodal vision-language models trained on billions of images. It predicts the 3D Gaussians from the point cloud input feed-forwardly, which are used for rendering later. 3D Gaussians are represented with xyz locations $\mathbb{R}^{N \times 3}$, covariance matrices $\mathbb{R}^{N \times 6}$, color matrices $\mathbb{R}^{N \times 3}$ and feature embeddings $\mathbb{R}^{N \times F}$. LIFT-GS predicts $\mathbf{G} \in \mathbb{R}^{N \times (m+12+512)}$ containing $m$ masks plus 12 channels for shape/location/color and 512 for feature loss:

$$\text{LIFT-GS: } (\mathbf{P}, \mathbf{Q}) \mapsto (\mathbf{G}, \mathbf{M}, \mathbf{C}), \quad (2)$$

In practice, training the masks using differentiable rendering adds only a small overhead during training. With the shapes of the inputs and outputs specified, the sections below describe each component: the losses used, pseudolabel generation, and the model architecture.

### 3.2. Losses

With differentiable rendering, LIFG-GS enables training 3D VLG models using the simplest 2D grounding losses. During training, LIFT-GS only requires (sparse) point cloud, 2D posed images as inputs without any other 3D annotations, which data can be easily obtained from RGB-D videos or SfM (Yang et al., 2025; Wang et al., 2025a; 2023; 2025b; Leroy et al., 2024) as done in (Szymanowicz et al., 2025).

LIFT-GS utilizes two groups of losses to train the model: $\mathcal{L}_{\text{ground}}$ for grounding (Jain et al., 2025), and per-pixel losses $\mathcal{L}_{\text{PP}}$ commonly used to improve results with differentiable

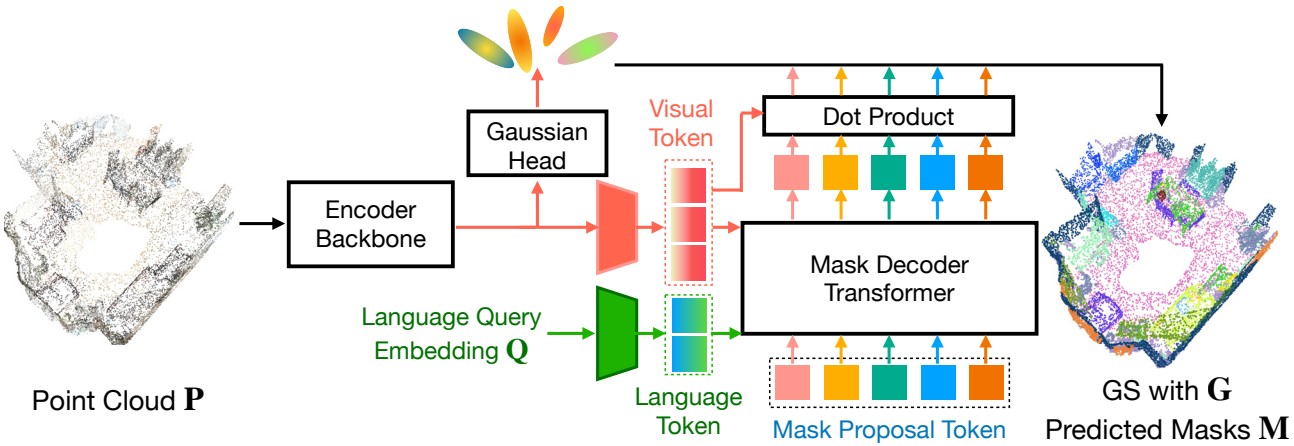

Figure 5: **Architecture Design**. LIFT-GS predicts 3D Gaussian Splatting $\mathbf{G}$ and 3D masks $\mathbf{M}$ given a point cloud $\mathbf{P}$ and language query embeddings $\mathbf{Q}$ as inputs. The 3D masks $\mathbf{M}$ are generated by a Transformer-based Mask Decoder.

rendering. For reference, rendered 2D masks, rgb images, and feature maps are denoted as $\tilde{\mathbf{M}}_{2D} \in \mathbb{R}^{H \times W \times m}$, $\tilde{\mathbf{F}}_{2D} \in \mathbb{R}^{H \times W \times F}$, and $\tilde{I} \in \mathbb{R}^{H \times W \times 3}$, respectively. Their corresponding ground-truth counterparts have analogous shapes: $\mathbf{M}_{2D}$, $\mathbf{F}_{2D}$, and $I$, where $\mathbf{M}_{2D} \in \mathbb{R}^{H \times W \times K}$.

### 3.2.1. *Grounding losses:*

LIFT-GS uses the MDETR-style mask grounding loss: $\mathcal{L}_{CE}$ and $\mathcal{L}_{mask}$ with $\mathbf{d}_{match}$ (matching distance). Since the number of predicted and ground-truth masks may differ, we apply Hungarian matching to pair them based on $\mathbf{d}_{match}$, and then optimize the matched predictions using $\mathcal{L}_{ground}$. We use the 2D variant of these losses under 2D rendering supervision, which can be easily replaced by their 3D counterpart (on 3D Gaussian centers) when 3D labels are available.

$$\mathcal{L}_{ground} = \frac{1}{K} \sum_i^K \lambda_3 \mathcal{L}_{mask}(\tilde{\mathbf{M}}_{2D}^{\sigma(i)}, \mathbf{M}_{2D}^i) + \lambda_4 \mathcal{L}_{CE}(\mathbf{C}_{\sigma(i)}, i) \quad (3)$$

$$\sigma(i) = \arg\min_j \mathbf{d}_{match}(\tilde{\mathbf{M}}, \mathbf{M}_i, \mathbf{C}) \quad (4)$$

**Cross Entropy Loss, $\mathcal{L}_{CE}$**: this loss supervises the correspondence between 3D mask candidates and input language tokens (*i.e.*, the matrix $\mathbf{C}$) by framing it as a (soft) classification problem. Recent work suggests that using BCE may produce sharper masks for long VLG text queries (Jain et al., 2025; Arnaud et al., 2025; Zhai et al., 2023), echoing similar findings in image segmentation (Cheng et al., 2021a).

$$\mathcal{L}_{CE}(\mathcal{C}_{\sigma(i)}, i) = -\log \frac{\exp(\mathbf{C}_{\sigma(i),i})}{\sum_j^K \exp(\mathbf{C}_{\sigma(i),j})} \quad (5)$$

**Mask loss, $\mathcal{L}_{mask}$**: this loss supervises the predicted 3D masks by comparing them to paired ground-truth 3D masks. Following SAM (Kirillov et al., 2023), we apply a combination of *Focal* (Lin et al., 2017) and *Dice* (Sudre et al., 2017) losses to supervise the predicted 3D masks effectively.

**Optimal matching, $\mathbf{d}_{match}$**: this function measures the pairwise distance between the 3D grounding results and the ground-truth values, and is used for matching. It is implemented similarly to $\mathcal{L}_{ground}$ but with different loss weights.

The model always predicts the maximum number of instances (256) but avoids false positive detections by matching the unused instance to a special `no-match` text token (Jain et al., 2021). The maximum of 256 was not a major limitation in practice, but could be increased.

### 3.2.2. *Per-Pixel Losses*

While grounding loss alone is sufficient for stable pretraining (Table 4), LIFT-GS benefits from joint training with additional photometric and feature losses for faster convergence. These losses are only used during pretraining, not for finetuning with 3D annotations.

**Reconstruction loss, $\mathcal{L}_{RGB}$**: supervises photometric reconstruction using $L_1$ and SSIM losses (Hong et al., 2024; Zhu et al., 2023b; Wang et al., 2004)

$$\mathcal{L}_{RGB} = \lambda_1 \mathcal{L}_1(I, \tilde{I}) + \lambda_2 \mathcal{L}_{SSIM}(I, \tilde{I}) \quad (6)$$

**Feature loss, $\mathcal{L}_{feat}$**: uses CLIP-style contrastive regularization to align rendered features $\tilde{\mathbf{F}}_{2D}$ with ground-truth features $\mathbf{F}_{2D}$ as in (Zhu et al., 2023b):

$$\mathcal{L}_{feat} = \frac{1}{H \times W} \sum_{u,v}^{H,W} -\log \frac{\exp(\tilde{\mathbf{f}}_{(u,v)} \cdot \mathbf{f}_k)}{\sum_j \exp(\tilde{\mathbf{f}}_{(u,v)} \cdot \mathbf{f}_j)} \quad (7)$$

where $\tilde{\mathbf{f}}_{(u,v)}$ is the rendered feature at pixel $(u,v)$ from $\tilde{\mathbf{F}}_{2D}$, $\mathbf{f}_k$ is the corresponding ground-truth feature, and $\mathbf{f}_j$ represents the batch of unique features.

## 3.3. Architecture

LIFT-GS is network-agnostic with minimal architectural constraints. The design can be applied to other architectures, with constraints only arising from the specific losses used. This section describes the network used in our experiments, shown in Figure 5.

### 3.3.1. *Grounding Decoder: (see Grounding Losses)*

The grounding decoder is a transformer based on Mask-Former (Cheng et al., 2021b) that predicts the correspondence matrix $\mathbf{C} \in \mathbb{R}^{m \times |Q|}$ and 3D Gaussian masks $\mathbf{M} \in \mathbb{R}^{N \times m}$. LIFT-GS uses the mask decoder from Uni-VLG (Jain et al., 2025) with minimal modifications.

The transformer takes Gaussian features $\mathbf{G}$ and language query embeddings $\mathbf{Q}$ as inputs, using a set of learnable tokens as 3D mask proposals. Cross-attention is computed between the learnable tokens and both the language and Gaussian tokens. After extensive information exchange within the transformer, $\mathbf{M}$ (or $\mathbf{C}$) is computed as dot products between mask proposal tokens and Gaussian tokens (or mask proposal tokens and language tokens, respectively). The detailed LIFT-GS architecture and hyperparameter settings used in our experiments are provided in Appendix A.

### 3.3.2. *Gaussian Decoder Head: (see Per-Pixel Losses)*

LIFT-GS predicts $\mathbf{G} \in \mathbb{R}^{N \times F}$ using a learned pointwise MLP applied to the point cloud encoder outputs. While the number of predicted Gaussians $|G|$ can differ from input points $|P|$, we set $|G| = |P|$ with a bijective mapping for consistency with point cloud evaluation tasks. Notably, the Gaussian decoder does not require direct supervision–3D Gaussians can be treated as latent variables for 3D VLG, as shown in Table 4 when per-pixel losses are disabled.

### 3.3.3. *Input Encoders:*

*Pointcloud Encoder*: Tokenizes RGB point clouds of shape $|P| \times 6$ (xyz + RGB) using a sparse convolutional UNet (Contributors, 2022), following PonderV2 (Zhu et al., 2023b). Weights are randomly initialized and learned.

*Text Encoder*: LIFT-GS uses CLIP text embeddings, which remain frozen during training.

These encoders represent common choices in 3D VLG. Stronger architectural choices like transformer-based pointcloud encoders or different text embeddings would likely improve performance.

## 3.4. SAM-CLIP 2D Pseudo-Label

While LIFT-GS eliminates the need for 3D annotations, obtaining high-quality 2D supervision remains challenging. We show one way in which 2D foundation models can gen-

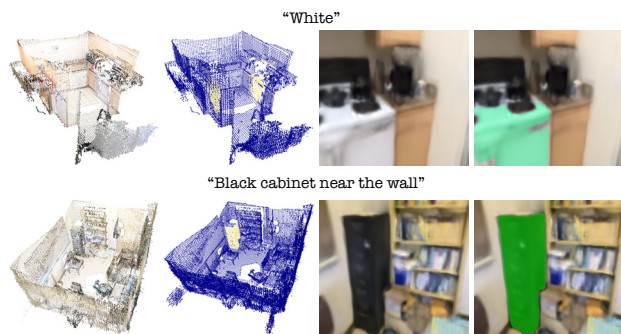

"White"

"Black cabinet near the wall"

Figure 6: **Zero-Shot 3D Segmentation.** Trained using only 2D pseudo-labels, LIFT-GS can localize objects in 3D from real text inputs in a zero-shot manner. From left to right, we visualize the *input point clouds*, *segmented 3D masks*(in yellow), *rendered images from predicted 3DGS*, and *rendered segmentation masks*. Language queries include both high-level abstract concepts (e.g., *white*) and detailed descriptions (e.g., *black cabinet near the wall*).

Table 1: **Open-Vocabulary 3D Instance Segmentation.** We evaluate our model on ScanNet200 by using category names as text queries and compare it against SOTA models.

| Model | mAP↑ | mAP25↑ | mAP50↑ |
|---|---|---|---|
| OpenScene (Peng et al., 2023) | 11.7 | 17.8 | 15.2 |
| OpenMask3D (Takmaz et al., 2023) | 15.4 | 23.1 | 19.9 |
| PQ3D (Zhu et al., 2024) | 20.2 | 32.5 | 28.0 |
| LIFT-GS-Scratch | 22.5 | 35.1 | 30.7 |
| LIFT-GS | **25.7** | **40.2** | **35.0** |
| Δ | +3.2 ↑ | +5.1 ↑ | +4.3 ↑ |

erate pseudo-labels that enable reasonable zero-shot performance and significantly enhance downstream fine-tuning.

As shown in Figure 4, we generate pseudo-labels using SAM (Kirillov et al., 2023) and CLIP (Radford et al., 2021). For each image, SAM provides segmentation masks, and for each segmented region, we extract CLIP image embeddings as *pseudo language query embeddings*. Since CLIP's text and image embeddings share the same feature space, LIFT-GS can use text embeddings during inference. We concatenate these CLIP embeddings to form $\mathbf{Q}$ and construct $\mathbf{C}$ s.t. each instance maps to exactly one query token.

With 2D pseudo-labels, LIFT-GS performs zero-shot 3D grounding using real text queries without fine-tuning (Figure 6). However, zero-shot performance suffers from low accuracy and struggles with complex expressions – a common limitation of CLIP-based methods that function as bag-of-words models (Yuksekgonul et al., 2023). Future improvements in pseudo-labeling, such as better captioning (Meta AI, 2024) and 2D language grounding models (Liu et al., 2023), could reduce reliance on fine-tuning.

LIFT-GS demonstrates that even simple pseudo-labeling strategies can be effectively distilled into 3D models. In the experiments below, pretraining with 2D pseudo-labels substantially improves downstream task performance. Fine-tuning data scaling results are consistent with established transfer learning "scaling laws" (Hernandez et al., 2021).

# 4. Experiments

Although supervised via 2D loss, LIFT-GS is a fully 3D model that explicitly outputs 3D masks. Beyond the zero-shot setting using 2D pseudo-labels (Figure 6), LIFT-GS can be readily fine-tuned with 3D annotation data using 3D losses for significantly improved performance. To enhance practical applicability, we focus on fine-tuning with 3D annotations and demonstrate how 2D model distillation boosts performance.

In this section, we first provide training details and show how pretraining significantly improves downstream task performance through a series of carefully designed ablations. We also reveal several insights from scaling the amount of pretraining and fine-tuning data, and explore the impact of different 2D foundation models.

## 4.1. Training Details

We provide details below with more details in the Appendix.

**Datasets** We use ScanNet (Dai et al., 2017) as the primary dataset for downstream task fine-tuning and evaluation, as its annotations form the basis for established benchmarks. We primarily pretrain using ScanNet (Dai et al., 2017) for comparison to other methods. LIFT-GS enables training on diverse unlabeled 3D datasets, and we show additional pretraining scaling experiments using ScanNet++(Yeshwanth et al., 2023), Taskonomy (Zamir et al., 2018) and Aria Synthetic (Somasundaram et al., 2023).

**Architecture** LIFT-GS method imposes minimal architectural constraints. In the following experiments, backbones consist of a text encoder (frozen CLIP-L), point cloud encoder (Sparse 3D UNet (Çiçek et al., 2016)), and grounding decoder (8-layer Transformer, hidden size 512 following (Jain et al., 2025)), and an MLP for the Gaussian decoder.

**Training** Models are trained end-to-end for 76k steps with batch size 32 on 32 A100s using AdamW (Loshchilov & Hutter, 2017) (lr=1e-4, weight decay=1e-4). Point clouds are voxel-downsampled to 5cm for an average of 50k points, and we render masks at resolution $512\times512$ to ensure small masks are captured. Complete implementation details are provided in the appendix.

## 4.2. Evaluation on 3D Vision-Language Grounding

We fine-tune and evaluate our pretrained model on two representative 3D VLG tasks: 3D open-vocabulary instance segmentation and 3D referential grounding. Our results, shown in Tables 1 and 2, demonstrate significant improvements over models trained from scratch and achieve state-of-the-art performance with pertaining.

### 4.2.1. GROUNDING SIMPLE NOUNS IN 3D

We first evaluate simple grounding for simple noun-phrases, using object categories without spatial relationships. Following the protocol in (Zhu et al., 2024), we convert the standard 3D instance segmentation benchmark on ScanNet into an open-vocabulary 3D instance segmentation task. The categories of objects are used as language queries, which are input to the model to predict the corresponding 3D masks.

**Evaluation setting:** We evaluate using the standard metric mAP, a measure of mask overlap averaged across categories. We fine-tune LIFT-GS for 500 epochs.

**Results:** Compared against the state-of-the-art baselines PQ3D (Zhu et al., 2024) and OpenMask3D (Takmaz et al., 2023), our pretrained model (LIFT-GS) achieves substantial performance gains (mAP 25.7% vs 20.2%), as shown in Table 1. It significantly outperforms its counterpart trained from scratch (LIFT-GS-Scratch mAP +3.2%).

### 4.2.2. GROUNDING COMPLEX PHRASES IN 3D

Next, we examine grounding multiple objects using more complex phrases that contain spatial references, referred to as *3D Referential Grounding* (3D RG).

**Evaluation Setting.** We evaluate LIFT-GS on the most common *3D Referential Grounding* benchmarks: ScanRefer (Chen et al., 2019), SR3D, and NR3D (Achlioptas et al., 2020; Abdelreheem et al., 2022). We use standard top-1 accuracy as the evaluation metric, considering a predicted bounding box correct if its IoU with the ground truth exceeds 0.25 or 0.5. Since LIFT-GS outputs masks instead of axis-aligned bounding boxes, we derive bounding boxes by extracting the extreme corner points from the point cloud within the predicted masks.

LIFT-GS is designed to be practical and we evaluate it using the "real-world" settings used in more recent 3D VLG work where (1) we predict 3D masks without assuming known ground-truth 3D bounding boxes, and (2) we utilize sensor point clouds (*Sensor PC*) from RGB-D scans instead of using mesh-derived point clouds that leak label information (*Mesh PC*). This realistic setting is more challenging, as reflected in the significant performance drop of BUTD-DETR (Jain et al., 2021) when transitioning from *Mesh PC* to *Sensor PC* (Table 2), consistent with findings in (Jain et al., 2024). A more complete comparison of these settings is provided in (Jain et al., 2021; 2025; Arnaud et al., 2025).

**Baselines** We compare LIFT-GS against the state-of-the-art two-stage methods, 3D-VisTA (Zhu et al., 2023c) and PQ3D (Zhu et al., 2024), as well as the SOTA single-stage method, BUTD-DETR (Jain et al., 2021). All two-stage baselines assume access to ground-truth 3D masks or boxes during inference, so we re-evaluate them using predicted

Table 2: **3D Referential Grounding.** We report top-1 accuracy with various IoU thresholds (0.25, 0.5).

| Method | SR3D | | NR3D | | ScanRefer | |
|---|---|---|---|---|---|---|
| | Acc@25 | Acc@50 | Acc@25 | Acc@50 | Acc@25 | Acc@50 |
| ***Mesh PC*** | | | | | | |
| LanguageRefer (Roh et al., 2021) | 39.5 | - | 28.6 | - | - | - |
| SAT-2D (Yang et al., 2021) | 35.4 | - | 31.7 | - | 44.5 | 30.1 |
| BUTD-DETR (Jain et al., 2021) | 52.1 | - | 43.3 | - | 52.2 | 39.8 |
| 3D-VisTA (Zhu et al., 2023c) | 56.5 | 51.5 | 47.7 | 42.2 | 51.0 | 46.2 |
| PQ3D (Zhu et al., 2024) | **62.0** | **55.9** | **52.2** | **45.0** | **56.7** | **51.8** |
| ***Sensor PC + Bounding Box Proposals using Mesh PC*** | | | | | | |
| 3D-VisTA (Zhu et al., 2023c) | 47.2 | 43.2 | 42.1 | 37.4 | 46.4 | 42.5 |
| ***Sensor PC*** | | | | | | |
| BUTD-DETR (Jain et al., 2021) | 43.3 | 28.9 | 32.2 | 19.4 | 42.2 | 27.9 |
| LIFT-GS-Scratch | 44.0 | 28.8 | 37.2 | 23.1 | 45.0 | 29.5 |
| LIFT-GS | **50.9** | **36.5** | **43.7** | **29.7** | **49.7** | **36.4** |
| Δ | +6.9(16%) | +7.7(27%) | +6.5(17%) | +6.6(29%) | +4.7(10%) | +6.9(23%) |

Table 3: **Comparison with other Pretraining Baseline.** LIFT-GS clearly outperforms Ponder-v2 and its variant Ponder-v2†, which is trained on the same SAM-CLIP features as ours.

| Model | Acc@0.25 | Acc@0.5 | Acc@0.75 |
|---|---|---|---|
| Scratch | 42.19 | 27.23 | 9.66 |
| Ponder-v2 (official) | 40.92 | 25.97 | 8.84 |
| Ponder-v2† | 45.40 | 29.36 | 9.29 |
| LIFT-GS | **47.53** | **33.75** | **13.49** |

Table 4: **Loss Ablation.** We show the impact of different pretraining losses on 3D referential grounding task. $\mathcal{L}_{ground}$ significantly improves results, particularly at high IoU thresholds.

| Model | $\mathcal{L}_{ground}$ | $\mathcal{L}_{RGB}$ | $\mathcal{L}_{feat}$ | Acc@0.25 | Acc@0.5 | Acc@0.75 |
|---|---|---|---|---|---|---|
| Scratch | | | | 42.19 | 27.23 | 9.66 |
| - | ✓ | | | 46.34 | 31.54 | 12.50 |
| - | ✓ | ✓ | | 46.67 | 31.81 | 12.45 |
| - | | ✓ | ✓ | **47.69** | 31.35 | 11.36 |
| - | ✓ | ✓ | ✓ | 47.53 | **33.75** | **13.49** |

boxes from the SOTA object detector Mask3D (Schult et al., 2022). For fairness, we re-train 3D-VisTA and BUTD-DETR on sensor point clouds. Because PQ3D uses multiple backbones and a multi-stage training pipeline, we were not able to reproduce PQ3D on the sensor point cloud setting.

**Results** Our model without pretraining (LIFT-GS-Scratch) achieves slightly better performance than the state-of-the-art single-stage method BUTD-DETR (Jain et al., 2021), likely due to architectural similarities with extra modifications.

With pretraining, LIFT-GS achieves significant improvements across all three datasets, with relative gains of $10\% - 30\%$, demonstrating the effectiveness of our pretraining approach. Notably, LIFT-GS outperforms 3D-VisTA in Acc@25, despite 3D-VisTA being a two-stage method with bounding box proposals from Mask3D using *Mesh PC*.

### 4.3. Pretraining Ablations

We conduct an in-depth analysis of the proposed method through a series of ablation and scaling experiments. For these evaluations, we use a model pretrained only on ScanNet as the baseline. To simplify the presentation for the ablations, we report results on the combined evaluation set of ScanRefer, SR3D, and NR3D. Additionally, we report the higher accuracy threshold Acc@0.75.

**Compare to SOTA pretraining methods** We compare

against PonderV2 (Zhu et al., 2023b), a state-of-the-art point cloud pretraining method that also uses render-supervision. Since the official PonderV2 relies on limited human-annotated text labels, we retrain it using our SAM-CLIP pseudo-labels for fair comparison (PonderV2† in Table 3). This demonstrates the value of distillation, improving the performance over using GT labels (45.4% vs 40.9% Acc@0.25). Moreover, LIFT-GS substantially outperforms PonderV2† (47.5% vs 45.4% Acc@0.25), underscoring the impact of multimodal decoder architectures enabled by the LIFT-GS render-supervised formulation.

**Loss Ablation** Existing pretraining pipelines primarily focus on the encoder (Zhu et al., 2023b; Banani et al., 2021), whereas the render-supervised formulation can pretrain the entire architecture in a unified manner using the grounding loss. We find that grounding loss alone can be used to pretrain the model end-to-end in Table 4. A model trained with $\mathcal{L}_{ground}$ alone (row 2) substantially improves downstream task performance, performing only slightly worse than the model trained with all losses (row 5). Furthermore, comparing models with and without $\mathcal{L}$ground (row 5 vs. row 4) clearly shows that $\mathcal{L}_{ground}$ significantly enhances downstream performance, particularly in more challenging scenarios (IoU thresholds of 0.5 and 0.75).

### 4.4. Data Scaling

LIFT-GS exhibits strong scaling properties that reveal 3D VLG operates in a severely data-scarce regime.

Table 5: **Fine-tune Data Scaling.** We show Acc@0.5 results with different ratio of fine-tuning data on referential grounding task.

| Finetuning Data Ratio | 10% | 20% | 50% | 100% |
|---|---|---|---|---|
| Scratch | 6.93 | 15.04 | 23.00 | 27.23 |
| LIFT-GS | 14.70 | 23.03 | 28.89 | 33.75 |

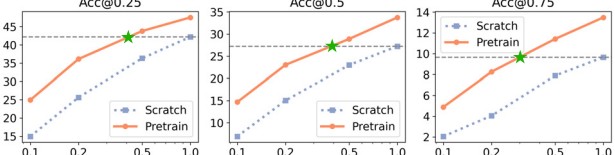

Figure 7: **Fine-tune Data Scaling.** We show how *Grounding Accuracy* changes with increasing *Data Ratio* from 0.1 to 1.0.

**Finetuning Data Scaling** We observe that pretraining effectively "multiplies" the fine-tuning dataset by approximately 2x. As shown in Figure 7 and Table 5, a pretrained model using 50% of fine-tuning data matches the performance of training from scratch with 100% data. This scaling coefficient remains constant across different data amounts (10%, 20%, and 50%) without diminishing–matching empirical scaling laws from other modalities (Hernandez et al., 2021). The benefits are most pronounced at higher IoU thresholds, where a pretrained model achieves scratch-level performance using only 30−40% of the fine-tuning data. Additional results on Instance Segmentation are provided in Appendix A.3.

**Pretraining Data Scaling** Expanding pretraining data consistently improves downstream performance (Table 6). Adding ScanNet++ yields notable gains (+0.8%), while incorporating Taskonomy and Aria Synthetic provides additional improvements despite distribution differences (likely due to the mesh reconstruction quality in Taskonomy).

Comparing the scaling results from pretraining and fine-tuning demonstrates the strong data efficiency of LIFT-GS. As shown in Table 6, adding ScanNet++ ( 30% of the data used for pretraining) yields a performance gain equivalent to adding 15% more ScanNet fine-tuning data with 3D annotations, based on the curve in Figure 7. This indicates an effective transfer ratio of roughly 1:2—*i.e.*, collecting twice as many raw videos provided improvements comparable to building a fully annotated 3D SFT dataset.

Therefore, pretraining on ScanNet++ is not only highly effective but also cost-efficient, especially considering that annotating 3D referential grounding data requires significantly more effort than collecting raw videos alone.

**Data Scarcity Implications** The consistent 2× multiplier without saturation, combined with continued gains from more pretraining data, strongly suggests that current 3D VLG models are severely limited by data availability–opening a path for future 3D VLG improvements through scaling alternate sources (such as 2D foundation models).

Table 6: **Pretraining on OOD data.** Adding more pretraining data from ScanNet++ improves performance. Taskonomy and Arial helped less than ScanNet++, likely due to distribution difference.

| Pretraining Data | Acc@0.25 | Acc@0.5 | Acc@0.75 |
|---|---|---|---|
| Scannet | 47.53 | 33.75 | 13.49 |
| +Scannet++ | 48.29 | 34.35 | 14.06 |
| ++ Taskonomy and Arial | 48.49 | 34.41 | 14.35 |

Table 7: **2D Foundation Model Exploration.**

| 2D Models | Acc@0.25 | Acc@0.5 | Acc@0.75 |
|---|---|---|---|
| SAM-B + CLIP-B | 46.31 | 31.50 | 12.41 |
| SAM-H + CLIP-L | 47.53 | 33.75 | 13.49 |
| SAM-H + LLAMA-Caption | 47.50 | 32.78 | 13.25 |

### 4.5. 2D Foundation Models Scaling and Exploration

Our pipeline leverages powerful 2D foundation models to generate pseudo-labels. Here, we investigate their impact by analyzing performance variations with different 2D foundation models, with results presented in Table 7.

**Weaker CLIP and SAM** The main experiments use SAM-H and CLIP-L for pseudo-labeling. Replacing them with smaller models, MobileSAM (Zhang et al., 2023a)(ViT-tiny) and CLIP-B, leads to a noticeable performance drop, especially at higher accuracy thresholds. This suggests that render-supervised distillation directly benefit from advancements in 2D foundation models.

**Captions from LMMs** Table 7 shows results using large LMMs instead of CLIP to generate queries (LLAMA-3V + SAM grounding, details in Appendix). After segmenting objects in 2D images using SAM, we prompt LLAMA-V to describe the segmented regions. Pretraining with these captions achieves performance comparable to our original pipeline with SAM-H and CLIP-L. As LMMs continue to improve, we believe text-based captions hold significant potential for future research, and the approach highlighted in the paper is positioned to benefit from LMM improvements.

## 5. Conclusion

LIFT-GS tackles data scarcity that limits 3D VLG by introducing render-supervised distillation from 2D VLM models. By training 3D models using only 2D supervision from models like SAM and CLIP, LIFT-GS achieves state-of-the-art performance for 3D VLG. Our findings, including consistent 2x data multiplication effects, reveal that 3D grounding currently operates with substantial data limitations. LIFT-GS circumvents a key 3D annotation bottleneck by introducing a scalable training approach that benefits directly from advancements in frontier multimodal language models. It offers a practical technique to leverage progress in 2D to accelerate the development of other data-scarce capabilities essential for robotics, AR/VR, and embodied AI.

## Acknowledgment

This work is done during Ang Cao's internship at Meta. We thank Andrew Owens, Andrea Vedaldi, Stella Yu, Ziyang Chen, Yiming Dou, Xuanchen Lu for their helpful discussion and feedback.

## Impact Statement

We propose a method for training 3D models without 3D supervision, advancing 3D vision-language research. Our approach significantly improves 3D referential grounding, a key task for robotics, embodied AI, and AR/VR applications. The resulting model enables agents to precisely localize objects from language inputs, bridging high-level reasoning with real-world actions.

As our model is distilled from 2D foundation models, it may inherit their biases. However, since our primary task is grounding, it is unlikely to introduce significant aesthetic biases.

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

# A. More Details

## A.1. Training Details

LIFT-GStakes point clouds and posed RGB images for training. For efficiency, we preprocess point clouds and posed RGB images, caching the processed features.

Point clouds originate from multi-frame RGB-D scans. We unproject them using depth information and fuse the unprojections into the final point clouds. Each dataset sample is preprocessed into 5cm-resolution point cloud chunks with corresponding posed RGB images.

For 2D pseudo-labels, we precompute SAM-CLIP features and cache them. Given the large size of the feature map, we decompose it into two components: *Semantics* and *Index2Semantics*.

*Semantics*: A tensor of shape $H \times W$, where each pixel stores the index of the segment it belongs to. *Index2Semantics*: A tensor of shape $N \times F$, where $N$ is the number of unique segments, and $F$ is the CLIP feature dimension. This decomposition significantly reduces storage costs. When computing the feature rendering loss $\mathcal{L}_{\text{feat}}$, we directly use features from *Index2Semantics* for contrastive loss.

Each training sample consists of a sparse point cloud and a posed image with corresponding SAM-CLIP features. We randomly sample up to 8 unique instances, using their CLIP features as *pseudo language queries* and their masks as target 2D masks. To ensure mask quality, we filter out masks smaller than 1024 pixels.

Randomly sampling instances is important for training, especially for zero-shot segmentation, as it prevents the model to reconstruct the whole images given all the input embeddings.

For grounding loss, we assign weights of 15.0, 2.0, and 6.0 to the mask cross-entropy loss, soft token loss, and Dice loss, respectively. We also use a photometric loss (L1 and SSIM) with a weight of 1.0 and a feature loss with a weight of 0.1.

UNet Encoder: 8 layers, maximum channel dimension of 256, output feature dimension of 96. MaskDecoder: 8-layer Transformer decoder with a hidden state size of 512. It uses 256 learnable mask proposal tokens, generating 256 masks. Each Transformer block has 8 attention heads, a feedforward MLP of dimension 2048, and a dropout ratio of 0.15. Language Encoder: We use `clip-vit-large-patch14`, with a feature dimension of 768.

## A.2. Comparison to 3D pseudolabels

Table 8: **Comparison to 3D pseudolabels.** A mask decoder trained on top of frozen LIFT-GS features matches and even outperforms a decoder trained on top of lifted 3D pseudolabels (voxel-pooled ConceptFusion (Jatavallabhula et al., 2023a)). LIFT-GS learns to pool features in 3D in order to optimally reproduce the pseudolabels after rendering, which outperforms using a hand-crafted aggregation. Note: in this experiment we used a more expressive mask decoder in this experiment with a larger MLP ratio, which improves the results for all methods, including LIFT-GS.

| Features | Acc@0.25 | Acc@0.5 |
|---|---|---|
| Scratch (RGB) | 44.1 | 30.6 |
| 3D pseudolabels | 50.1 | 34.7 |
| 2D pseudolabels (LIFT-GS features) | 51.8 | 38.3 |
| LIFT-GS (finetuned) | 54.7 | 40.5 |

## A.3. Data Scaling Results

A similar trend is observed for 3D open-vocabulary instance segmentation, though the benefits of pretraining are slightly less pronounced due to the task's lower complexity. This aligns with our findings that pretraining is more beneficial for challenging tasks, such as those with higher IoU thresholds or greater complexity.

## A.4. VLM Captions

We explore using vision-language models (VLMs) to generate captions for each SAM-segmented object and encode these captions into CLIP embeddings as *pseudo language queries*.

Specifically, given a SAM-segmented region, we draw a red bounding box on the 2D image and highlight the masked region

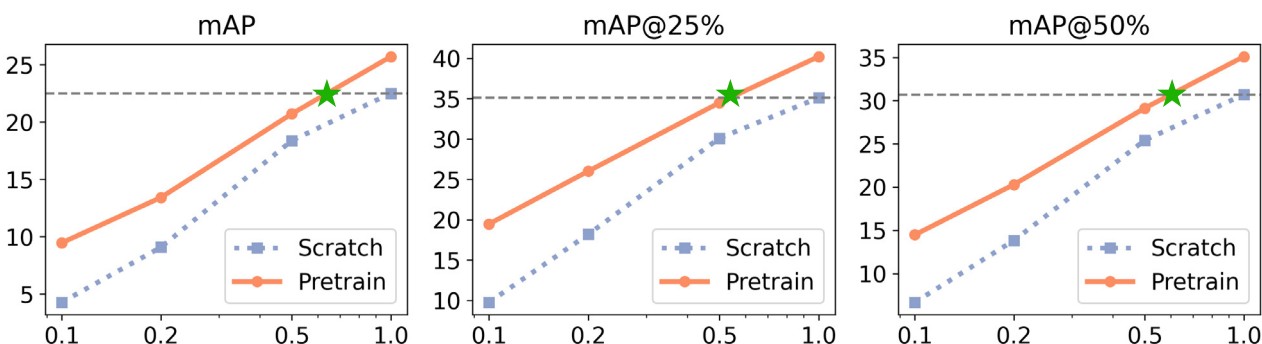

Figure 8: **Finetunning Data Scaling on Open Vocabulary 3D Instance Segmentation.** We show how *mAP* changes along with increasing *Data Ratio* from 0.1 to 1.0

using alpha blending, as shown in Figure 9. We then prompt a VLM, such as LLama-3.2v, with the following instruction:

You are a helpful assistant for image captioning. You are given an image with a red bounding box specifying the object of interest. Caption that object in a few words, keeping it precise and concise. The object is also slightly highlighted. Examples output: "a red traffic light," "the box near the wall." Just output the caption; no other text is needed.

This approach leverages VLM-generated textual descriptions to improve pseudo-language queries for training.

## B. Discussion and Limitations

The core contribution of LIFT-GS is training a 3D model without 3D supervision by leveraging differentiable rendering and distilling knowledge from 2D foundation models. This approach is novel and motivated by the fact that 2D foundation models, trained on vast amounts of 2D data, currently outperform any existing 3D model. Distilling knowledge from these powerful 2D models presents a promising and scalable direction for 3D learning.

Our proposed pipeline is general and unified. Beyond 3D masks, any renderable 3D attributes can, in principle, be trained using 2D supervision. This idea could extend to dynamic scenes and other properties, opening new opportunities for 3D model training.

However, LIFT-GS is inherently constrained by how well we leverage 2D foundation models for pseudo-labeling. Currently, we use CLIP image embeddings as text queries, but CLIP's claim of a shared embedding space for images and text is imperfect. In practice, these embeddings can differ significantly, leading to challenges in zero-shot 3D segmentation.

Our CLIP-SAM features may not be optimal pseudo-labels for pretraining, and we anticipate that improved pseudo-labeling strategies will lead to better scaling properties, stronger performance, and even robust zero-shot 3D segmentation without fine-tuning. Addressing our current limitations presents a key opportunity for future work.

Although LIFT-GS significantly improves performance and surpasses the single-stage SOTA method BUTD-DETR (Jain et al., 2021), it still falls short of two-stage SOTA methods like 3D-VisTA (Zhu et al., 2024) on 3D referential grounding at an IoU threshold of 0.5. A robust single-stage 3D VLG model would have a major impact across various applications. We hope that our architecture-agnostic pretraining pipeline can further enhance future models.

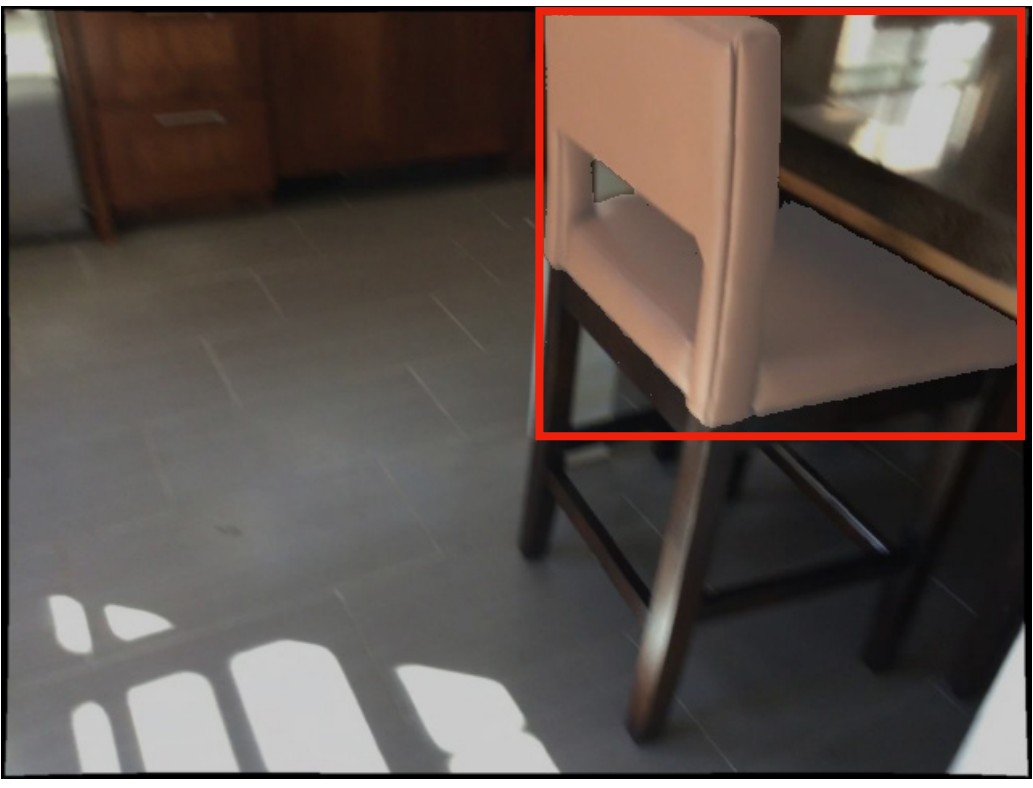

Figure 9: **Input Image to VLM for Captions.** Given the segments from SAM, we draw red bounding box around the segments and ask VLM models to describe the segments inside the red bounding boxes.

