# OpenReview forum: "From Thousands to Billions: 3D Visual Language Grounding via Render-Supervised Distillation from 2D VLMs"
_ICML.cc/2025/Conference — ICML 2025 poster_

### Official Review · Reviewer_35FE · 2025-03-13

**Overall Recommendation:** 2

**Summary:**

This paper proposes an approach for 3D vision-language understanding by leveraging rendered RGB images, grounding masks, and 2D feature loss for model training, rather than incorporating explicit 3D supervision. The model follows a pretrain-finetune paradigm, with evaluations conducted on open-vocab 3D instance segmentation and 3D referential grounding. Ablation studies demonstrate that the pretraining stage enhances fine-tuning efficiency by reducing the amount of required data.

## Post rebuttal
I appreciate the authors’ clarifications regarding the model architecture, pretraining strategy, and the paper’s claims. Although this direction is promising, I cannot currently recommend acceptance due to several issues:
(1) the omission of critical technical details in the methods section;
(2) numerous typos and grammatical errors;
(3) overstatements of the model’s capabilities; and
(4) insufficient explanation as to why pretraining on ScanNet is beneficial compared to ScanNet++. Instead, the authors only compare pretraining on ScanNet++ followed by fine-tuning on ScanNet.

Concerns (1) and (2) were also raised by Reviewer eYEH, and (3) was flagged by Reviewer zvKK. The authors acknowledge these points in the rebuttal. In the paper’s current form, there is more confusion than clarity: for instance, the model details remain unclear, it’s not included which data the model is pretrained on, how exactly that pretraining is conducted, and why ablations and data scaling are evaluated on a combined set of ScanRefer, SR3D, and NR3D. Furthermore, the remaining grammatical errors and overstated claims may mislead the readers, and I encourage the authors to revise and polish the paper to improve its coherence.

**Claims And Evidence:**

- The paper proposes a framework for pretraining and fine-tuning on 3D VL tasks, and its performance is verified by the experiments and ablations.

- However, "The approach is new for vision-language understanding." (Line 16) is not accurate as using rendered features/images as supervision is not new in 3D VL understanding, as also pointed out by the paper's related work section, e.g., Ponderv2, Point Cloud Unsupervised Pre-training via 3D Gaussian Splatting (Liu et all.). This paradigm is the first in the 3D referential domain, to the best of my knowledge.

- The claims in the introduction "render supervised framework can be used with essentially any 3D/4D task or model, provided the results are renderable" is quite bold to make. For example, extending the proposed method to tasks requiring image inputs (e.g., fine-grained caption) instead of sparse point clouds or to 4D dynamic scenes is inherently non-trivial.

**Essential References Not Discussed:**

Not that I'm aware of.

**Experimental Designs Or Analyses:**

The experiment results generally validate the effectiveness of the proposed model. I have two questions:

1) In Tab.4 Loss Ablation, what is the performance when only L_{RGB} is excluded? It seems the L_{RGB} loss plays a negligible role in the performance.

2) Could the author provide insights into why adding Scannet++ only marginally improves the performance in Tab. 6? ScanNet++ dataset, while smaller than ScanNet, is roughly 1/3 of the ScanNet scenes. Is using half of the ScanNet scenes realizing similar performance, similar to the data scaling of the fine-tuning data?

**Methods And Evaluation Criteria:**

The paper lacks sufficient details of the proposed methodology, including both the model and data perspectives.

- The paper lacks a detailed explanation of the model architecture and its operational flow. While it specifies the inputs and outputs of each module, it does not clarify how inputs are processed within the model or how outputs, such as the correspondence matrix C, are subsequently utilized. Furthermore, Figure 4 and the accompanying text are not well-aligned, leading to significant confusion for readers trying to understand the overall framework.

- The paper does not mention how (much) pretraining data is generated or used in the pretraining. Are all the images used for each scene? Is the sensor PC organized by the RGBD point clouds for all the frames? Then how to handle the increasingly large point clouds as the model input.

**Other Comments Or Suggestions:**

Typos, for example, on Line 22-23 "For training, only need images and camera pose, and 2D labels.", Line 25-26 "We demonstrate this to pretrain a network", Line 90 "Specifically, render supervised framework can be used".

**Other Strengths And Weaknesses:**

The writing of the paper could be further refined. Currently, it contains typographical errors, unjustified hypotheses in the introduction, and a method section that lacks sufficient details.

**Questions For Authors:**

While I appreciate the results demonstrated in the experiment section, as well as the data scaling performance in the ablation, I cannot give a positive rating at the moment primarily due to my concerns over the method and pretraining data as outlined above. Especially, how do the readers comprehend the effectiveness of pretraining when the details of pretraining data are not presented? The authors are encouraged to address my concerns above.

**Relation To Broader Scientific Literature:**

I believe the proposed method aims to address the scarcity of data in 3D vision language understanding domain, by leveraging the benefits of neural rendering, e.g., Gaussian Splatting, and its strong relationship with point clouds. The paper presents evidence of improving 3D referential grounding through this paradigm, which I believe can bring potential insights for the community.

**Theoretical Claims:**

No theoretical claims in the paper.

---

> ### Author Rebuttal · Authors · 2025-03-31
>
> We appreciate that the reviewer likes our results and data scaling performance. We answer the questions below and will improve the writing given the valuable suggestions.
>
> ---
>
> > Model architecture and its operational flow…
>
> **Encoder Backbone**:  is a SparseConv UNet [1] (Sec. 3.4), following PonderV2 implementation.
>
> **Gaussian Head** is a lightweight MLP (2 layer, 128 dim)  mapping each point from the Encoder Backbone to Gaussian parameters (L182-L190), including scaling, color, etc.
>
> The **Mask Decoder Transformer** follows the MaskFormer pipeline [2], using m learnable tokens (mask proposal tokens in Figure 4) to predict potential 3D masks. Each token corresponds to a binary 3D instance mask over the Gaussians.
>
> The **correspondence matrix** C (m x |Q|) grounds the m instance masks into the Q language tokens. Each element indicates the probability that the mask corresponds to that particular language token.
>
> The **mask decoder** is a Transformer decoder, where the visual and language tokens serve as the Key and Value, while the proposed mask tokens act as the Query inputs to the decoder.
>
> To further aid clarity and reproducibility, we will release the code and improve the final version. Lastly, we note that LIFT-GS is model-agnostic, imposing minimal architectural constraints and being readily adaptable to other architectures (Lines 224–230).
>
>
> > Pretraining Data
>
> We used the training scenes from ScanNet and ScanNet++ for pretraining, except for the ablation study in Table 6. For images, we sampled frames from the original video trajectories with a frame skip of 30 (i.e., at 1 Hz). Each selected RGB-D frame was unprojected using the provided camera intrinsics and poses, then voxel-pooled at a 5 cm resolution, so the total point cloud number is controlled.
>
>
> ### Claims
>
> > Using rendered features/images as supervision is not new in 3D VL understanding... This paradigm is the first in the 3D referential domain, to the best of my knowledge.
>
> We agree and will make the claim more precise by specifying that we mean grounding with complex language (3D referential grounding).
>
>
>
> > The claim "render supervised framework…" is quite bold to make. For example, extending... or to 4D dynamic scenes is inherently non-trivial.
>
> We appreciate the reviewer’s suggestion and will revise it to make it more accurate.  We appreciate any recommendations, and suggest the following: “The render-supervised framework provides a general and extensible design. LIFT-GS shows how to use it for highly structured tasks, such as 3D referential grounding and object detection”.
>
>  We genuinely believe that our pipeline presents a general and extensible design. While adapting it to new tasks may involve non-trivial effort, we argue that such extensions are both feasible and conceptually straightforward.
>
> For image inputs, one could use methods like Dust3R [3] to regress point maps from images as point clouds. For dynamic scenes, it is possible to regress motion basis coefficients for each point, as Shape-of-Motion[4]. These examples illustrate our belief that the proposed pipeline can be extended to support a wide range of applications beyond the current scope.
>
>
> ### Experimental analysis
>
> > Role of L_{RGB} loss
>
> We keep L_{RGB} loss because it is necessary to supervise the reconstruction of the 3D Gaussian fields. We agree that this loss may not significantly benefit downstream tasks, as it provides limited semantic information.
>
> > Improvement of adding Scannet++ in Table6?
>
> Excellent point. In Tab. 5, increasing the finetuning data by 100% yielded about a 5%  Acc@0.5 improvement. Increasing the pretraining data by 30% yielded a 1% improvement.
>
> Based on the curve in Figure 6 (we show the real and estimated values based on the curve below), the performance gain from adding ScanNet++ data (30% more) for pretraining is roughly equivalent to adding 15% more ScanNet finetuning data with 3D annotations.  It shows that the effective transfer ratio is roughly 1 / 2; i.e., collecting twice the number of raw videos alone can yield improvements comparable to building a fully annotated 3D dataset.
>
> Therefore, **pretraining on ScanNet++ is highly efficient and cost-effective**, considering that annotating 3D referential grounding data requires orders of magnitude more effort than collecting raw video alone.
>
> |Finetune Data Ratio| 10%   | 20%    | 50%  | 100% | 115%| 130%  | Pretraining on Scannet++(130%)|
> |-|-|-|-|-|-|-|-|
> | Acc@0.25             | 24.96  | 36.11  | 43.80 | 47.53 | 48.28 |  48.94| 48.29|
> | Acc@0.5              |  14.70  | 23.03 | 28.89  | 33.75 | 34.72 | 35.59 | 34.35|
> | Acc@0.75            |  4.89    | 8.26   | 11.42  | 13.49 | 13.91 | 14.27 | 14.06|
>
> [1] https://github.com/facebookresearch/SparseConvNet/blob/main/examples/ScanNet/unet.py
>
> [2] MaskFormer: Per-Pixel Classification is Not All You Need for Semantic Segmentation
>
> [3] DUSt3R: Geometric 3D Vision Made Easy
>
> [4] Shape of Motion: 4D Reconstruction from a Single Video

---

### Official Review · Reviewer_aEzm · 2025-03-13

**Overall Recommendation:** 4

**Summary:**

The work addresses the problem of open vocabulary based 3d segmentation, that is predicting 3d masks for an RGB point cloud that adhere to the language based query.
In order to do so the authors propose a feedforward architecture that predicts 3D Gaussians which carry information about their belonging to segmentation masks.  These masks are predicted via a transformer decoder from the language query and the encoded point cloud (with learnt mask tokens similar to the decoder of MaskFormer).
A key point of the paper is the supervision via 2D pseudo ground truth (with is plenty) rather than 3d labels.  This is achieved by rendering the Gaussians and their masks and comparing them against 2d segmentations built from SAM masks and corresponding CLIP language embeddings.  The model is optimized on the (i) RGB reconstruction, (ii) mask and language consistency, and (iii) a feature loss.
Experimental results demonstrate the effectiveness of the approach for open vocab instance segmentation (ScanNet200) and 3D referential grounding (ScanRefer, SR3D, NR3D).  Further ablation studies provide insights into the losses, the impact of pretraining, and the importance of good 2D foundational models for pseudo ground truth generation.

**Claims And Evidence:**

The main claim of the paper is to distilling knowledge from 2D foundation models for supervision rather than relying on 3d labels.  This argument is well accepted in the community and underlined by the scarcity of 3d labels.

**Essential References Not Discussed:**

Liu et al., "Weakly Supervised 3D Open-vocabulary Segmentation", NeurIPS'23

Qin et al., "LangSplat: 3D Language Gaussian Splatting", CVPR'24

**Experimental Designs Or Analyses:**

A comparison between PQ3D and the proposed approach is missing on the very same input data.  The authors argue that they didn't succeed in retraining PQ3D on "Sensor PC" input data.  Though, LIFT-GS could have been trained and evaluated on "Mesh PC" data.  Even though this input data might not be available in real-world applications, it would allow for an apples-to-apples comparison to the SoTA method.  Given the present analysis it appears that PQ3D might outperform the proposed method.
A comparison would provide clarity. However, the novelty of the approach lies within the generic (pre)training from 2d ground truth rather than beating the top performer on any benchmark.

**Methods And Evaluation Criteria:**

The method is well presented and builds upon existing building blocks (differentiable rendering via Gaussian splatting, mask prediction via learnt proposal tokens, SAM based segmentation, CLIP for language embeddings) but also the usage of 2d ground truth has been well explored in the community (which is appropriately cited in the related work section).

**Other Comments Or Suggestions:**

N/A

**Other Strengths And Weaknesses:**

N/A

**Questions For Authors:**

- How are the different scales of maks that are generated by SAM handled in the loss?  Segmentations in different images that see the same object will contradict such that a consistent 3d segmentation is not easily possible. He. et al. "View-Consistent Hierarchical 3D Segmentation Using Ultrametric Feature Fields", ECCV'24 addresses this problem via a hierarchical segmentation.

- Which model is used to compute the ground truth feature map F_2D?

**Relation To Broader Scientific Literature:**

Open vocabulary 3d instance segmentation is a very active research field.  As such the proposed 3d supervised pre-training task is valuable to the wider community.

**Theoretical Claims:**

N/A

---

> ### Author Rebuttal · Authors · 2025-03-31
>
> We sincerely appreciate the reviewer’s recognition of LIFG-GS, including “the method is well presented and builds upon existing building blocks”,  and believing  “As such the proposed 3d supervised pre-training task is valuable to the wider community.”. We answer the question below and will make them more clear in the final version.
>
> ---
> >How are the different scales of maks that are generated by SAM handled in the loss? Segmentations in different images that see the same object will contradict such that a consistent 3d segmentation is not easily possible. He. et al. "View-Consistent Hierarchical 3D Segmentation Using Ultrametric Feature Fields", ECCV'24 addresses this problem via a hierarchical segmentation.
>
> That’s an excellent question, and we appreciate the opportunity to elaborate on this point. A key advantage of our approach is that it does **not** require 3D view-consistent masks, as our method is fully learning-based and relies solely on 2D supervision.
>
> In contrast, existing methods that leverage SAM often rely on hand-crafted hierarchical structures or empirically designed mask-merging heuristics to improve consistency across views. These procedures are typically complex and heuristic-driven, introducing additional sources of noise and making the pipeline less robust.
>
> Our framework takes a different route: the 3D model is queried using 2D CLIP embeddings, and training is guided only by corresponding 2D mask and grounding losses. As a result, there is no requirement for the 2D masks to be consistent across views. Instead, we encourage the Transformer decoder to **learn how to produce coherent 3D masks** that align with these independently supervised 2D views, in a fully data-driven manner—without relying on manually designed post-processing steps.
>
> To extract masks from SAM, we sample a grid of points across each image and generate masks accordingly. We apply Non-Maximum Suppression (NMS) to reduce overlapping masks and discard extremely small ones. Aside from these minimal filtering steps, we do not apply any further merging or heuristic post-processing.
>
> ---
>  >Which model is used to compute the ground truth feature map F_2D?
>
> We use the same pipeline as in Figure 3 to compute the ground-truth 2D feature map F_2D​. Specifically, we apply the CLIP image encoder to extract features for each segmented region, assigning the resulting feature vector to all pixels within that region. For segmentation, we adopt the SAM-H model, and we use the CLIP-L model to extract features. This pipeline is flexible and can also accommodate alternative backbones, such as DINO-v2.
>
> ---
> >A comparison between PQ3D and the proposed approach is missing on the very same input data…. However, the novelty of the approach lies within the generic (pre)training from 2d ground truth rather than beating the top performer on any benchmark.
>
>
> We sincerely appreciate the reviewer’s recognition that the novelty of our work lies in proposing a generic (pre)training framework using 2D supervision, rather than merely achieving state-of-the-art results on specific benchmarks.
>
>
> Regarding PQ3D, we note that the reason why our method does not use exactly the same input data—i.e., mesh point clouds—as PQ3D,  is that mesh point clouds are impractical in real-world applications due to the need for time-consuming meshing and human annotations (L298-303). To ensure a fair comparison, we made every effort to retrain all baselines under our unified setting for an apples-to-apples comparison. However, we were unable to successfully reproduce PQ3D due to its complex, multi-stage training pipeline that spans across multiple datasets.
>
> We greatly appreciate the reviewer’s suggestions and agree that more rigorous, apples-to-apples evaluations—using identical input settings as prior works—would further improve the clarity and fairness of our comparisons. While we are actively working on this, the time and computational constraints during the rebuttal period prevent us from including those results here. We plan to incorporate them in a future revision of the paper.
>
> ---
> > Additional Reference
>
> Thanks for pointing out, and we will include them in the final version.

---

### Official Review · Reviewer_eYEH · 2025-03-13

**Overall Recommendation:** 3

**Summary:**

This paper presents LIFT-GS, a scalable pretraining approach for 3D vision language grounding. Specifically, the model takes in a point cloud of the scene along with the language query embeddings to produce 3D Gaussians with features together with the predicted masks for grounding. For training LIFT-GS, the reconstruction loss on RGB, feature loss on image features from 2D foundation models, and the grounding loss on the predicted masks are applied. The method does not require ground truth 3D annotations or even 2D annotations. Instead, it leverage 2D foundation models to generate pseudo-labels for training. Experimental results on 3D language grounding demonstrate the decent the performance of the proposed method.

## Update after rebuttal

My concerns about comparing with more recent state-of-the-arts and the training cost comparison are mostly solved. However, since the high-level idea is somehow similar to LangSplat which makes the Gaussians to predict other features and properties beyond the original properties, I would regard the method as a good direction to try on 3D language grounding, but the concept itself is not super novel. Therefore, I would like to maintain my score of weak accept.

**Claims And Evidence:**

Yes, the claim that the proposed LIFT-GS acts as a pretraining solution for 3D language grounding without 3D supervision is supported by the experiments.

**Essential References Not Discussed:**

There might be some more recent works that can be used for comparison. For example, SceneVerse [1] and D-LISA [2]. Also, LangSplat [3] should be discussed in the literature as it is the first to enable open-vocabulary language grounding in 3D scenes with Gaussian Splatting, although LangSplat is per-scene optimized instead of training across scenes in a feed-forward way as the proposed method. Other than that, there are no essential references that are not discussed.


[1] Jia et al. SceneVerse: Scaling 3D Vision-Language Learning for Grounded Scene Understanding. ECCV 2024.

[2] Zhang et al. Multi-Object 3D Grounding with Dynamic Modules and Language-Informed Spatial Attention. NeurIPS 2024.

[3] Qin et al. LangSplat: 3D Language Gaussian Splatting. CVPR 2024.

**Experimental Designs Or Analyses:**

Yes, the experimental designs or analyses are sound and valid, except that I have some concerns about whether there are stronger and more recent state-of-the-arts for comparison. I will elaborate on this point in the "Other Strengths And Weaknesses" section.

**Methods And Evaluation Criteria:**

Yes, the proposed methods and the evaluation criteria make sense.

**Other Comments Or Suggestions:**

There are some typos and strange grammars in some parts of the paper. I list some of them here:
- Line 22: "For training, only need images and camera pose, and 2D labels." The sentence is weird.

- Line 25-26: "We demonstrate this to pretrain a network ..." The expression of this sentence is strange.

- Line 163: "ganulariry" -> "granularity"

- Line 182: "Lmask" -> "$L_{\rm mask}$"

- Line 205: "$C\sigma(i)$" -> "$C_{\sigma(i)}$"

The correspondence matrix $C$ in Equation 1 has not be explained until Lines 194-195, which makes me feel very confused when I first read the paper.

**Other Strengths And Weaknesses:**

**Strengths:**
- The proposed method achieves training a 3D vision language grounding model with no ground truth 3D supervision. Moreover, the 2D labels can also be pseudo-labels generated by 2D foundation models. It can be very useful for scaling up the training data as the training process does not require 3D annotations.
- The proposed training pipeline of cross-scene render-supervision is innovative to me in the context of 3D vision language grounding. Nevertheless, I have to admit that both (1) cross-scene 3D Gaussians / neural rendering and (2) language embedding in 3D Gaussians with additional features are not completely new ideas. For (1), the related works include pixelSplat [1], MVSplat [2], etc. For (2), the related works include LangSplat [3].


**Weaknesses:**
- For the experimental comparisons, the most recent state-of-the-art for fair comparison seems to be 3D-VisTA [4], which seems to be a relatively out-of-date work for me. I think there are more recent advancements for 3D vision language grounding, including SceneVerse [5], D-LISA [6], etc. The authors seem to miss on discussing these related works, and therefore missing the experimental comparisons with them.
- The training cost seems to be huge, which requires 32 A100 GPUs. Compared to other methods like SceneVerse [5] (which is already a method for scaling up data) which only requires 8 A100 GPUs for 2 days, the training cost of the proposed method seems to be very intimidating.

[1] Charatan et al. pixelSplat: 3D Gaussian Splats from Image Pairs for Scalable Generalizable 3D Reconstruction. CVPR 2024.

[2] Chen et al. MVSplat: Efficient 3D Gaussian Splatting from Sparse Multi-View Images. ECCV 2024.

[3] Qin et al. LangSplat: 3D Language Gaussian Splatting. CVPR 2024.

[4] Zhu et al. 3D-VisTA: Pre-trained Transformer for 3D Vision and Text Alignment. ICCV 2023.

[5] Jia et al. SceneVerse: Scaling 3D Vision-Language Learning for Grounded Scene Understanding. ECCV 2024.

[6] Zhang et al. Multi-Object 3D Grounding with Dynamic Modules and Language-Informed Spatial Attention. NeurIPS 2024.

**Questions For Authors:**

- How is the proposed method compared with the more recent state-of-the-arts like SceneVerse [1] and D-LISA [2]? Are there any specific reasons for the authors not discussing these works and not comparing with them in the experiment section? If not, I think comparing with
- What is the training cost comparison between the proposed method and the compared baselines? The paper only states that the method needs to run 76K steps on 32 A100 GPUs. However, there are no specific training time / cost comparison between LIFT-GS and the other baselines.


[1] Jia et al. SceneVerse: Scaling 3D Vision-Language Learning for Grounded Scene Understanding. ECCV 2024.

[2] Zhang et al. Multi-Object 3D Grounding with Dynamic Modules and Language-Informed Spatial Attention. NeurIPS 2024.

**Relation To Broader Scientific Literature:**

The paper is focusing on 3D vision language grounding, which is broadly related to 3D multi-modal language models. It can be a primitive step towards artificial general intelligence.

**Theoretical Claims:**

The paper does not have any proofs or theoretical claims.

---

> ### Author Rebuttal · Authors · 2025-03-31
>
> We thank the reviewer for their thoughtful feedback and appreciate the recognition that using 2D supervision is “very useful for scaling up the training data,”  “the claim… is supported by the experiments.”,  “cross-scene render-supervision is innovative…”. We address the questions below and will incorporate improvements in the final version.
>
> ---
> > Compared to baselines, which output language embedding in 3D Gaussians with additional features, like Langsplat.
>
> Thanks for the suggestion — we will clarify it in the final version. Both LIFT-GS and LangSplat use render-supervised distillation. However, LangSplat's per-scene optimization method is primarily designed for object segmentation (nouns only), while LIFT-GS is a cross-scene method supporting 3D referential grounding, involving more complex language.
>
> LangSplat and its variants rely on **language-vision feature dot products for open-vocabulary segmentation**: reconstructing 3D RGB and CLIP feature fields and computing dot products with text embedding for segmentation (**Encoder only**). However, contrastive language-vision models (e.g., CLIP) tend to behave like bag-of-words models [2], making them struggle with handling slightly long language expressions with relational structure, such as spatial relationships, key in referential grounding.
>
> In contrast, LIFT-GS employs structured supervision from MDETR, training a **transformer decoder** over visual and language tokens to directly predict 3D masks and groundings through learned attention and referential loss.
> **We illustrate the failure modes of dot-product-based methods on both  3D and 2D with language inputs in Fig** https://postimg.cc/bd6QKTYk, https://postimg.cc/47Y9ZCxc.
>
> We compare LIFT-GS and LangSplat variants on 3D referential grounding benchmark ScanRefer. As LangSplat operates on rendered 2D images instead of 3D, we use a variant Semantic Gaussians [1] that reports much higher performance and directly segments 3D; as well as LERF (which both papers compare to).
>
> | Method | Acc@0.1 | Acc@0.25  | Acc@0.5|
> |-----------|---------------|-----------------|-------------|
> |  Semantic Gaussians |  18.2% |  8.2% | 3.0%|
> | LERF | - | 4.4% | 0.3% |
> | LIFT-GS | - | 49.7% | 36.4%|
>
>
> ---
> > Comparison and discussion about SceneVerse [5], D-LISA [6].
>
> Thanks for your suggestions, and we will include comparisons and discussions in the final version.
> The submission incudes comparisons to PQ3D (ECCV 2024), which reports stronger performance than both SceneVerse and D-LISA. For open-vocabulary segmentation, we compare against the numbers reported in PQ3D and show that LIFT-GS outperforms it.
>
> For the 3D referential grounding task, we tried our best efforts to retrain baselines under the Sensor Point Cloud setting for a fair comparison; adopting PQ3D, 3D VISTA, and BUTD-DeTr as our main baselines. Despite our best efforts, we weren't able to train PQ3D. However, we believe that 3D-VISTA is the appropriate comparison here, since the requested references perform comparably to that method on mesh point clouds. We compare their reported results as published below.
>
> Finally, we emphasize that LIFT-GS is both **model- and data-agnostic**, making it orthogonal to advances in data (e.g., SceneVerse) or architecture (e.g., D-LISA). Stronger models or larger 3D datasets can be seamlessly incorporated into our framework for better performance, as indicated in Figure 6.
>
> | Method       | SR 0.5@Acc-Multiple | SR 0.5@Acc-Unique|  Contribution|
> |-----------------|-------------------------------|----------------------------| ----------------|
> | PQ3D         |46.2| 78.2 | - |
> | 3D-VisTA  | 39.1| 75.1 | -|
> | D-LISA      | 40 | 75.5 | Architecture|
> | SceneVerse| 42.7| 77.9| New Data|
>
> ---
> > Computation Resource Comparison.
>
> The difference in computational cost primarily arises from the **type of point cloud data** used during training. We follow the setting of BUTD-DeTR, using sensor point clouds—unprojected from posed RGB-D frames (frame skip of 30, voxel size 0.05 cm)—resulting in ~30k points per scene. This setup reflects real-world use cases more closely (L305).
>
> In contrast, most other methods,  including SceneVerse, use ScanNet mesh segments, which are pre-annotated via face clustering, reducing scenes to ~300–1500 segments (∼100× fewer elements). This significantly lowers computational cost.
>
> However, meshing is not viable for real-world or real-time applications—it is computationally expensive and also contains human-annotated information. That said, for fairness, LIFT-GS can also operate on mesh segments, and under this setting, training requires comparable compute (8×A100 GPUs for 2–3 days).
>
> ---
> > Typos:
>
> We greatly appreciate these catches and will fix them in the final version.
>
> [1] Semantic Gaussians: Open-Vocabulary Scene Understanding with 3D Gaussian Splatting
>
> [2] [When and Why Vision-Language Models Behave like Bags-Of-Words, and What to Do About It?](https://openreview.net/forum?id=KRLUvxh8uaX)

---

> > ### Comment · Reviewer_eYEH · 2025-04-07
> >
> > Dear authors,
> >
> > Thanks for providing the rebuttal! My concerns about comparing with more recent state-of-the-arts and the training cost comparison are mostly solved. However, since the high-level idea is somehow similar to LangSplat which makes the Gaussians to predict other features and properties beyond the original properties, I would regard the method as a good direction to try on 3D language grounding, but the concept itself is not super novel. Therefore, I would like to maintain my score of weak accept.

---

### Official Review · Reviewer_zvKK · 2025-03-14

**Overall Recommendation:** 2

**Summary:**

The paper presents LIFT-GS, a feedforward 3D vision–language grounding model that accepts a point cloud and a language query as inputs. It converts the point cloud into 3D Gaussians and uses differentiable rendering to supervise training with only 2D losses. The system is distilled from 2D foundation models to achieve 3D mask predictions for language-described objects.

**Claims And Evidence:**

- Supervision Without 3D Labels

This is not a novel concept nor a new problem; similar ideas have been explored via NeRF-based per-scene optimization (LERF, NeRF-DFF), Gaussian splatting (LangSplat), feedforward transformers (Large Spatial Model, SAB3R), and even 3D LLMs (3D-LLM). Moreover, the choice to use point clouds rather than multi-view images appears as a mere setting difference. It remains unclear why the authors focus exclusively on point clouds.

- Technical Novelty

The method sections (3.1–3.4) describe a task formulation, standard 2D-supervised training for 3D reconstruction, and a conventional network for lifting point clouds into 3D Gaussians.
The contributions in terms of problem and model novelty are not sufficiently distinguished from prior work. A stronger claim on what specific challenges are uniquely solved by focusing on point clouds and 3D Gaussian representations is needed.

**Essential References Not Discussed:**

While the references are adequate, the paper does not make a clear case for a distinct problem being solved or a novel method proposed. A more critical comparison to similar recent methods would help clarify its contributions.

**Experimental Designs Or Analyses:**

Clarify how the model’s performance scales with data volume (how many scenes are used) and whether the absence of RGB images (as inputs) impacts the supervision via differentiable rendering.

**Methods And Evaluation Criteria:**

- Input Modality and Framework Choice

The paper focuses solely on point cloud input. Since differentiable rendering with known camera poses is well established for RGB images, the choice to restrict the input to point clouds raises questions.  It would strengthen the work to clarify why point clouds are preferred in this context and to discuss how the framework might extend (or why it might not extend) to settings with multiple images.

**Other Comments Or Suggestions:**

The authors should clarify why focusing on point clouds is beneficial and what new challenges are addressed by their specific combination of 3D Gaussian representations and 2D supervision. Strengthening the discussion on how their approach differs from existing feedforward or per-scene optimization methods would improve the paper’s impact.

**Other Strengths And Weaknesses:**

### Strengths:

- The paper leverages a scalable pipeline using differentiable rendering and distillation from 2D foundation models.
- It demonstrates state-of-the-art performance on downstream 3D vision–language grounding tasks.

### Weaknesses:

- The choice of point clouds over multi-view images is not sufficiently justified.
- The overall novelty in both problem setting and technical method is not clearly established beyond what prior works have already addressed.

**Questions For Authors:**

See the questions above.

**Relation To Broader Scientific Literature:**

The paper’s positioning relative to prior 3D reconstruction and vision–language grounding work should be discussed in greater depth. The literature shows many similar approaches; hence, a clear delineation of the unique aspects of this work is essential.

**Theoretical Claims:**

The manuscript does not propose any novel theoretical claims; its contributions are primarily empirical and architectural.

---

> ### Author Rebuttal · Authors · 2025-03-31
>
> First, we would like to thank the reviewer for taking the time and effort to engage with the paper, and we look forward to a productive discussion. Before addressing the individual questions, we would like to clarify the core focus of our work.
>
> While our method involves 3D scene reconstruction as an intermediate step, the primary goal of this paper is **3D vision-language grounding**—specifically, localizing 3D instances (i.e., point cloud masks) based on complex language queries. 3D spatial understanding is a known weakness of existing VLMs ([1; Fig. 5]), but is a critical capability for applications across robotics and embodied AI. For example, it can identify pick-and-place locations in long-horizon manipulation and rearrangement tasks.
>
> The key challenges and contributions of our work lie in developing an effective and scalable pipeline for (pre-)training large 3D visual-language grounding models (the **Decoder**), and the contribution is to do this using differentiable rendering using 2D supervision (Line 97).
>
> ---
> > similar ideas have been explored via … feedforward transformers (Large Spatial Model, SAB3R)
>
> Could the reviewer kindly provide a citation or reference for SAB3R? Despite our best efforts, we have been unable to find a corresponding paper or preprint. On the project website, we found [3], the paper link on the project website returns a 404. The only reference we found is on a personal website [2], which does not link to any publication or arXiv submission.
> Moreover, none of the other referenced papers evaluate on any of the standard 3D visual-language grounding benchmarks (ScanRefer, SR3D, NR3D).
>
> ---
> > Why point cloud inputs?... why focusing on point clouds is beneficial
>
> As mentioned in the paper, the losses used in LIFT-GS are largely agnostic to inputs and could work with multi-view image inputs. The fact that the introduced problem formulation is independent of the input type highlights the primary focus of our paper, which is **a new method for training 3D visual-language grounding models (especially Decoder)**, and we believe this is a strength of the method.
>
> We chose to focus primarily on point cloud inputs, since they are currently one of the most widely adopted 3D representations in robotics, and the return type of SLAM and SfM. Point clouds offer flexibility and modularity, as they decouple the 3D reconstruction process from the grounding task, allowing us to leverage a wide range of input sources—including single-view, sparse-view, or long RGB(-D) videos, as well as LiDAR scans or other sensors. This makes point cloud-based methods especially well-suited to diverse real-world conditions.
>
> Since pointcloud reconstructions can be incrementally updated and are largely independent of the specific sensor package (camera, depth sensor, IMU, etc). These properties make point clouds a practical choice for real-world robotic systems. For better or worse, they have become a de facto representation for 3D language grounding.
> We will make this point much clearer in the final version.
>
> ---
>
> > Technical Novelty and compared to prior works. The method sections (3.1–3.4) describe a task formulation, standard 2D-supervised training for 3D reconstruction, and a conventional network for lifting point clouds into 3D Gaussians. The contributions in terms of problem and model novelty are not sufficiently distinguished from prior work. A stronger claim on what specific challenges are uniquely solved by focusing on point clouds and 3D Gaussian representations is needed.
>
> Reconstructing 3D Gaussian splats from point clouds is neither the focus nor a claimed contribution of our work. Our primary contribution lies in **(pre-)training a large Transformer-based mask decoder for 3D visual-language grounding without requiring 3D labels**, made possible through the use of differentiable rendering.
>
> In contrast, the mentioned prior methods—whether based on per-scene optimization or feedforward models—focus on reconstructing the **RGB and feature fields** of the 3D scene and perform language grounding by computing **dot products between language CLIP features and 3D feature fields**. These approaches primarily consist of encoder-only architectures, whereas our work emphasizes the use of a powerful Transformer-based decoder for mask prediction. We discuss it in L96-109, L153-164.
>
> Due to the limitations of CLIP embeddings, dot-product-based grounding methods struggle with handling **slightly complex language expressions** and cannot reliably **reason about relative spatial relationships**. This restricts their effectiveness in realistic grounding tasks. Because of character limits, we provide quantitative comparisons and discussions in the reply for Reviewer eYEH.
>
> [1] https://open-eqa.github.io/
>
> [2] https://tianx-ia.github.io/
>
> [3] https://web.archive.org/web/20250501000000*/https://uva-computer-vision-lab.github.io/sab3r/static/pdf/Semantic_Augmented_3D_Foundation_Models__Writing.pdf

---

### Decision · Program_Chairs · 2025-05-01

**Decision:**

Accept (poster)

**Comment:**

After the review and rebuttal, two reviewers recommend acceptance, two rejection. The authors resolved most issues. The paper seems to be of general interest, and the AC has sufficient confidence that the authors can (and will) address the few remaining concerns in a revision to the paper:

As summarized by Reviewer 35FE

> (1) the omission of critical technical details in the methods section;
>
> (2) numerous typos and grammatical errors;
>
> (3) overstatements of the model’s capabilities; and
>
> (4) insufficient explanation as to why pretraining on ScanNet is beneficial compared to ScanNet++. Instead, the authors only compare pretraining on ScanNet++ followed by fine-tuning on ScanNet.